# Epithelial to Mesenchymal Transition: A Mechanism that Fuels Cancer Radio/Chemoresistance

**DOI:** 10.3390/cells9020428

**Published:** 2020-02-12

**Authors:** József Dudás, Andrea Ladányi, Julia Ingruber, Teresa Bernadette Steinbichler, Herbert Riechelmann

**Affiliations:** 1Department of Otorhinolaryngology and Head and Neck Surgery, Medical University of Innsbruck, 6020 Innsbruck, Austria; julia.ingruber@i-med.ac.at (J.I.); teresa.steinbichler@i-med.ac.at (T.B.S.); herbert.riechelmann@i-med.ac.at (H.R.); 2Department of Surgical and Molecular Pathology, National Institute of Oncology, 1122 Budapest, Hungary; ladanyi@oncol.hu

**Keywords:** silibinin, MRX34, PD-L1, NRF2, Krüppel-like factors (KLFs), neurotrophin

## Abstract

Epithelial to mesenchymal transition (EMT) contributes to tumor progression, cancer cell invasion, and therapy resistance. EMT is regulated by transcription factors such as the protein products of the SNAI gene family, which inhibits the expression of epithelial genes. Several signaling pathways, such as TGF-beta1, IL-6, Akt, and Erk1/2, trigger EMT responses. Besides regulatory transcription factors, RNA molecules without protein translation, micro RNAs, and long non-coding RNAs also assist in the initialization of the EMT gene cluster. A challenging novel aspect of EMT research is the investigation of the interplay between tumor microenvironments and EMT. Several microenvironmental factors, including fibroblasts and myofibroblasts, as well as inflammatory, immune, and endothelial cells, induce EMT in tumor cells. EMT tumor cells change their adverse microenvironment into a tumor friendly neighborhood, loaded with stromal regulatory T cells, exhausted CD8^+^ T cells, and M2 (protumor) macrophages. Several EMT inhibitory mechanisms are instrumental in reversing EMT or targeting EMT cells. Currently, these mechanisms are also significant for clinical use.

## 1. Introduction

Epithelial and mesenchymal cells are two major cell types. However, trans-differentiations of epithelium into mesenchyme (EMT) and of mesenchyme into epithelium (MET) occur during embryonic development [1]. The reversible program of the trans-differentiations between the epithelial and mesenchymal endpoints is crucial for embryonic development. Importantly, both directions of trans-differentiation are reactivated in many cancer types, but a full transition from an epithelial starting point into a differentiated fibroblastic endpoint with the expression of a fibroblast surface protein or vimentin is rarely executed [2,3]. The EMT typical for cancer cells is incomplete and is characterized by the induction of EMT-transcription factors (EMT-TFs), which increase cancer cell motility, allowing either the dissemination of individual tumor cells or the collective migration of cell clusters [2]. Nevertheless, the EMT-TFs play even more important pleiotropic roles [4] in invasive, disseminating, and progressive cancer. Their most important role is in maintaining stemness properties, as recent reports link EMT-TFs to cancer stem cells [2,5]. Moreover, EMT-TFs are also activated in non-epithelial tumors, such as leukemia [6].

The requirement for EMT in the route from a primary tumor to metastasis is debated, but most authors agree that tumor cells require plasticity, which allows back and forth switches between epithelial and mesenchymal phenotypes to adapt themselves to different hostile conditions [2,7].

During the epithelial to mesenchymal transition, epithelial cells undergo morphological changes, redirect their apical-basal cell polarity toward a front-rear polarity, give up their epithelial differentiation, gene expression profile, and morphology, release their lateral cell junctions and their connections to the basal substrate, and elongate and acquire motile and invasive properties. This is a reversible transition, which is reverted by MET [3].

The publications of Elisabeth Hay were the first to highlight these transition processes [8] in embryonic development, organ pathologies, and tumor cell metastasis [9]. In 2005, Elisabeth Hay, together with D. LaGamba and A. Nawshad [10], investigated the rapid morphological changes in a developing mouse palate, where they isolated the medial edge epithelium, without contamination of the surrounding mesenchymal cells. The morphological changes were a loss of cell–cell adhesion, an elongation of the cells, and an invasion of the underlying extracellular matrix of the new, transformed, mesenchymal cells. In this work, the authors indicated that epithelial cells from the medial edge epithelium trans-differentiate into newly-formed mesenchymal cells, which migrate through the extracellular matrix to specific locations associated with their developmental programs [10].

Epithelial cells interact with matrix components on their basal surface via receptors, which also interact with the basal actin cortex inside the cells. In contrast, mesenchymal cells interact with the extracellular matrix all around their circumference [3]. These cells also move by continuously constructing a new front-end, and the myosin-rich endoplasm slides into the renewing front-end [3]. During EMT, scattered cells emigrate from the basal surface-attached epithelium by turning on the front-end migration mechanism of the mesenchymal cells. These cells move into the matrix, and their whole circumference comes in contact with the extracellular matrix [9]. At the same time, inside the EMT cells, the basal actin cortex is reorganized into bundles of stress fibers [3].

In addition to the precise description of the morphology changes in newly developing mesenchymal cells of epithelial origin, the studies of Elisabeth Hay on embryo development indicated the involvement of WNT-signaling in EMT and, after that, the role of transforming growth factor-beta (TGF-β) in causing EMT in both development and pathology [11].

EMT is not only a key element in embryonic development and organogenesis [12], but it has been identified as a probable response to organ damage and a loss of functional epithelial cells [13,14,15,16]. In this context, the participation of tubular epithelial cells (TEC) in kidney fibrosis should first be mentioned. During kidney fibrosis, TECs trans-differentiate into collagen-producing cells, which are phenotypically similar to myofibroblasts [17]. Several years later [18], the EMT hypothesis was also proposed in relation to the liver, which suggests that hepatocytes may develop a (myo)fibroblastic phenotype and contribute to fibrogenesis [13]. In the last 20 years, it was found that, both in relation to the kidney and the liver, the EMT hypothesis is debatable [13,15]. The controversy resides not in whether the EMT hypothesis is correct or incorrect, but in its extent and its importance. Even though the EMT hypothesis in relation to fibrosis is widely accepted among authors in the field, its contribution to the generation of the extracellular matrix producing myofibroblasts is considered to be relatively little (according to some studies, only up to 5%) [15].

The hypothesis that mesenchymal trans-differentiation of epithelial cancer cells supplies tumor cells with a significant invasive ability as well as with an active extracellular matrix. Matrix-degrading-enzymes are dated earlier than the hypothesis that it has a role in fibrosis [1,8,9,19]. EMT presents itself as a reversible process in organogenesis and cancer progression, which empowers tumor cells with the plasticity required for invasion and therapy resistance [12]. This reversibility constitutes an appealing approach in the design of new cancer therapy pipelines. Drug discovery aims to induce EMT reversal, such as the recently proposed vorinostat [12], which is a histone deacetylase inhibitor (HDACi). The essential goal of EMT reversal remains the reactivation of the epithelial differentiation pattern in tumor cells that have undergone a mesenchymal transition. At the same time, the therapeutic activation of epithelial re-differentiation might contribute to an improved epithelial seeding of metastatic cell colonies and nodules and should be considered carefully [2].

This review briefly summarizes the EMT regulatory mechanisms in cancer, which elucidates the mechanistic connection between EMT and radio/chemoresistance introduces the novel aspect of EMT and immune checkpoint interplay, and concludes with a focus on developing cancer therapeutic options that target EMT.

This review article is based on our own original results [20,21,22,23,24,25,26], which have been published elsewhere and on a broad database that has been systematically reviewed to consider the main topics in this work (see Appendix A). The review concentrates on the recent development of the EMT hypothesis in both solid tumors and leukemia. The latter is rarely discussed in relation to EMT. A very important point of view is the discussion on the skeptical aspect of EMT [2,27,28,29]. The clinical significance of EMT is also an important aspect. How well can EMT be demonstrated in patient-derived tissue material? Is this mechanism a relevant target for cancer treatment? These questions are also addressed in this review article.

## 2. EMT Mechanisms, Its Interactions with Tumor Microenvironment and Its Targeting

### 2.1. EMT-Regulating Mechanisms

Despite the continuously accumulating knowledge on the mechanisms of EMT, it is not completely clear how this process is regulated. This lack of understanding has induced skepticism regarding the significance of the EMT hypothesis in relation to the tumor metastasis process, which was first discussed by David Tarin in 2005 [27] and placed into context in a comprehensive review article by Nieto et al. [29]. The essential principle that EMT takes place in ‘real cancers’ has been seriously doubted by David Tarin, whose review article in 2005 indicates, to state it simply, that EMT-like mechanisms can be recorded in vitro, but in vivo evidence of the existence of EMT in carcinoma progression is difficult to obtain [27]. One option to obtain better proof is the identification of EMT cells in tumor tissue sections by selective, reliable, and sensitive immunolabelling [30]. Another option is the use of genetically engineered mouse models that are suitable for the investigation of the consequences of a loss of genes expected to play major roles in the EMT processes [2]. The genetic deletion of SNAI1 and TWIST1 EMT-TFs in a pancreatic cancer model revealed that the loss of these EMT-TFs did not reduce metastasis but increased chemosensitivity [2,28,30]. In contrast, if ZEB1 was deleted in an identical pancreatic cancer model, metastasis was reduced by 40% [31]. Moreover, if SNAI1 was deleted in a genetically engineered breast cancer model, metastasis was impaired without compensation [32]. These newer findings argue for at least a partial dispensability of EMT-TFs for the metastatic mechanism, but phenotypic plasticity still seems to be necessary for metastasis [2,33]. Nevertheless, a new report describes the dissemination of squamous cell carcinoma cells, attached to tumor fibroblasts, as epithelial–mesenchymal couples, which allows epithelial and not mesenchymal transdifferentiated cells to enter the metastasis process [28,34]. Genetic engineering techniques in cancer models addressed the doubts regarding the role of EMT-TFs in metastasis and showed that some EMT-TFs are dispensable in this relation [30,31,32,34].

While EMT is inducible in cultured tumor cells in vitro [2], its clinical relevance for ‘real-life’ clinical settings has been questioned and debated [2,27]. The main problem is that we cannot perform a lineage-tracking of the genetic fate-mapping of cells in patient cancer tissue. Therefore, immunolabelling techniques of EMT markers in cancer tissue sections constitute the only way to estimate the proportion of EMT cells [2,35,36]. The proportion of the EMT cell population in both kidney fibrosis [15] and in cancer [30,36] is 5%, which indicates that EMT is a focal event [37]. Following a complete EMT conversion of epithelial tumor cells into mesenchymal ones, the resulting cells cannot be distinguished from connective tissue cells, and they share their markers [2]. In tumor cells of metastatic nodules, the previous history of the EMT program also cannot be revealed [30].

Taken together, in vitro studies provided exciting mechanisms of EMT, while the extent of EMT cells in tumor tissue, due to the insensitivity of the detection methods, is low, and the EMT-TFs are not clearly associated with metastasis (several of them are even dispensable) [2]. Nevertheless, these debated points are becoming better clarified both by the improvement of cancer models and by a more sensitive complex immunolabelling of heterogenic cancer cells in tumor tissue. Our research group is making important efforts to resolve the latter issue [21,23,38].

The potential discrepancy between the experimental mechanistic data and the clinical relevance of EMT is the major contradiction in this connection. Our own experimental data [20,23,24,25,26], in agreement with the reports of other authors, suggest that EMT might be a coincidental interplay of external initiating factors and intracellular supporting metabolic conditions [39]. The exogenous factors inducing EMT include chemical carcinogens, viruses, radiation, hypoxia, and an acidic microenvironment. The microenvironment surrounding the tumor cells is rich in cells producing growth factors that promote EMT. The interplay between microenvironmental inducing components and intracellular conditions enabling a response activates signal transduction pathways such as Ras, Myc, Bmi-1, Oct4, Nanog, Slug, Twist, Zeb1, and Zeb2 [7]. In addition to the activation of transcription factors by signal transduction, microRNAs (miRNAs) [40,41,42] are also involved in EMT regulation. EMT, in turn, is a powerful mechanism by which an originally unreceptive cancer surrounding develops into a cancer-friendly microenvironment [7,40] (Figure 1).

The EMT mediator described by Elisabeth Hay and her colleagues is TGF-β1 [46]. TGF-β1 is regarded as a double-edged sword during tumorigenesis [37], which reflects its ability to be a tumor suppressor in normal tissue and early-stage cancers and also a tumor promoter in progressive cancer [46]. There are higher TGF-β1 concentrations near the tumor vasculature, and the cancer cell nest contains a decreasing TGF-β1 gradient from its border toward the middle. This gradient generates heterogeneity between the tumor–stroma interface and the center of the tumor cell nest [47] (Figure 2).

The TGF-β1 gradient contributes to the heterogenic architecture of the tumor cell nest, where the invasive front has a completely different functional and differentiation profile from the core of the tumor cell nest. Nieto et al. addressed this architecture in a comprehensive review article, published in 2016 [29]. The polarization of cancer cells through TGF-β1 results in a slower-cycling TGF-β1-responsive population at the border of the cancer cell nests and in a non-TGF-β1-responsive population, which proliferates faster and accelerates tumor growth. The TGF-β1-responding cells are the ones that invade neighboring tissues, show differentiation changes, and activate the EMT gene expression program. At the same time, the TGF-β1-responding EMT cancer cells are radio-resistant and chemo-resistant [42,49,50,51]. This latter issue might be due to the TGF-β1-dependent transcriptional induction of p21, which stabilizes the nuclear factor erythroid-2-related factor 2 (NRF2) and leads to an enhancement of glutathione metabolism. This has the potential to weaken anti-cancer therapeutics [47]. The new findings on TGF-β1–EMT research harmonized the experimental data with the architecture of tumor cell nests and helped to elucidate the importance of EMT in ´real cancers.´

TGF-β1 connects epithelial solid tumors with leukemia in terms of EMT. An EMT-like process in leukemia has also been shown in a published study [52]. The bone marrow microenvironment is comparable to the tumor microenvironment surrounding the cancer cell nests of solid tumors containing scattered EMT cells (Figure 2). Both are hypoxic niches that produce TGF-β1 [52]. Novel reports argue that the bone marrow microenvironment might be responsible for refractory acute myeloid leukemia (AML) [53]. Targeting the intrinsic resistance of tumor cells did not improve the clinical outcomes of AML [53]. A new hypothesis argues for an EMT-like change in leukemia cells, which also includes the induction of EMT transcription factors, such as ZEB2 [2,54,55]. In AML, the leukemic blasts detach from the bone marrow niches and move into the peripheral blood to colonize other sites by changing the adhesion conditions [6], which is very similar to the adhesion changes described for EMT in the Introduction [6]. The E-cadherin promoter is hypermethylated in leukemia, which leads to the loss of this protein [56], as in the tumor cells of solid tumors, which can lose E-cadherin by down-regulating EMT [24]. EMT is not only a mechanism to acquire a motile phenotype, but it is also associated with the maintenance of the cancer [5]. Similar to solid tumors, in leukemia, EMT-TFs, such as TWIST, ZEB1-2, and SNAI1-2, play critical roles [6,55]. For example, the knockdown of ZEB1 in a leukemia model drastically reduced the blast invasion [57]. These recent findings indicate that EMT is not only a process of the mesenchymal transition of epithelial tumor cells in solid tumors, but leukemia cells also activate EMT transcription factors to attain invasion and therapy refractory properties.

Increasingly, scholars are arguing that the most important event in EMT is the induction of EMT-TFs, which play important roles in solid tumors as well as in leukemia [2,5]. EMT is associated with a core transcription signature, characterized by the down-regulation of E-cadherin and the up-regulation of SNAIL, ZEB, and TWIST TFs (Figure 1). The detection of these TFs in cancer tissue is negatively correlated with survival [2,58]. These TFs show pleiotropic effects, and their importance is beyond the coordination of the replacement of the epithelial phenotype with a mesenchymal one [2]. For example, Slug (SNAI2), which is a member of the SNAI EMT-TF family, functions as a master regulator and allows cells to enter into the tumor-initiating cell state [59]. The other member of the SNAI family, called Snail1, is mostly a strong transcriptional repressor, and it also represses Prrx1 TF [40]. Prrx1, in turn, is able to repress Snail1, but not directly. The activation of the microRNA miR-15f is required for this action. Both Snail1 and Prrx1 are induced by TGF-β1. Snail1 is an early response gene and Prrx1 is activated later. The chronological order is as follows: Snail is activated first, and, as a strong repressor, it reduces the lateral adhesion of epithelial tumor cells. Then, it allows the epithelial tumor cells to be released from their neighbors. Subsequently, Prrx1, which is a strong mesenchymal inducer, is activated, which promotes the acquisition and maintenance of mesenchymal features and allows for cell migration. The mesenchymal cells are able to migrate to the metastatic location, but Prrx1 must be down-regulated again during the course of MET in order to allow the docked metastatic cells to colonize at the secondary localization [40]. This is just an example of an elegant investigation of a fine-tuned mechanism, described in 2019, which hides behind the complex regulation of the plasticity of the tumor cells in relation to EMT and MET.

In addition to the main stream of SNAIL, ZEB, and TWIST TFs, an interesting group of transcription factors that regulate epithelial or mesenchymal gene expression profiles and contribute to the epithelial or mesenchymal phenotype is the group of zinc finger transcription factors: the Krüppel-like factors (KLFs). KLF4, for example, is associated with epithelial differentiation and is down-regulated by TGF-β1 during EMT [60]. In contrast, KLF7 is a positive regulator of EMT. Its over-expression contributes to the migration activity of oral squamous cell carcinoma (OSCC) cells, and it induces EMT and lymph node metastasis by activating Snail [61]. According to Nieto and colleagues, Snail plays a critical role in the inhibition of the E-cadherin gene expression [62,63,64,65]. Snail directly binds to responsive E-box sequences in the E-cadherin (CDH1) promoter [65]. Subsequently, enzymes belonging to repressor complexes are recruited to initiate histone modifications and DNA methylation, which leads to an entire modification of the chromatin structure. The consequence is a direct suppression of the tight junction and gap junction proteins between epithelial cells [66].

In addition to the transcription factors, the EMT-related gene expression changes are regulated by non-coding RNAs, such as long noncoding RNAs (lncRNAs) and microRNAs (Figure 1). A newly identified lncRNA, termed the metabolism-induced tumor activator 1 (MITA1), is induced by metabolic stress and contributes to metastasis [67]. MITA1 induces the transcription factor, Slug, and, in this indirect way, it also promotes EMT [67]. The finding that MITA1 also regulates EMT, in addition to external factors, such as TGF-β1, is a significant step in EMT research and offers an insight into the complexity of EMT regulation through the interaction of inner and external contributors [67].

An interesting external determinant of EMT is the acidic microenvironment, which is a consequence of tumor cell metabolism. This is dominated by aerobic glycolysis [68]. In an acidic condition, tumor cells tend to release exosomes. These small membrane-covered vesicles are considered to be intercellular communicators, which might be associated with tumor progression, recurrence, and metastasis. The exosomes contain microRNAs that are released in the acidic microenvironment and contribute to cell proliferation, migration, and the invasion of recipient tumor cells. The most important functional miRNAs, released from the exosomes in an acidic microenvironment, are miR-21 and miR-10b. Moreover, the acidic microenvironment also triggers the activation of hypoxia-related transcription factors, such as HIF-1α and HIF-2α, which stimulate exosomal expression and the release of miR-21 and miR-10b [69]. In a self-activating acidification (hypoxia), an exosome release circle is built up in a microenvironment, which results in the accumulation of miR-21 and miR-10b.

The altered metabolism in tumor cell nests results not only in an acidic microenvironment, but also in hypoxia (Figure 1 and Figure 2) [68]. The effects of a hypoxic microenvironment are cellular metabolism alterations and modified molecular responses to triggers [70]. Hypoxia and tumor progression in concert with radiation-resistance and chemoresistance are considered to be a joint profile of tumors. Hypoxia is induced by its main transcriptional activators known as the hypoxia-inducible factors (HIFs). The genes regulated by HIFs are seen as a cluster, which are involved in cell survival, proliferation, motility, metabolism, pH regulation, extracellular matrix function, inflammatory cell recruitment, and angiogenesis. Hypoxia also triggers EMT in several types of cancer, including breast, prostate, and oral cancer [70]. The connection between hypoxia, HIFs, and EMT is the previously mentioned transcription factor: Slug. Slug is extremely elevated in head and neck squamous cell carcinoma (HNSCC) cells in response to hypoxia/HIF-1α overexpression [71], which is also associated with a cadherin switch, the risk of lymph node metastasis, and a more advanced TNM stage [71]. In conclusion, the association of Slug and HIF seems to be proven in HNSCC and the data of Zhang and colleagues suggest that Slug and HIF-1 are potential risk markers predicting the clinical outcome of HNSCC [71].

### 2.2. Interaction with an Inflammatory Microenvironment

Tumor-associated chronic inflammation is a hallmark of cancer [72], which fosters the progression to metastasis [37]. EMT tumor cells and infiltrating inflammatory cells establish a bidirectional cross-talk that may result in a phenotype switch of inflammatory cells into tumor-supporting cells. This has been exemplified by the ‘M2’ polarization of macrophages, which is referred to as tumor-associated and tumor supportive [73]. The M2-polarization means that cancer cells induce the up-regulation of CD163 and Arg1 and the down-regulation of IL-1β and IL-6. Feng and colleagues describe this process in the following way: cancer cells educate macrophages, which, in turn, activate Nrf2 in cancer cells [74]. This interaction might also be responsible for increasing EMT in cancer cells through paracrine VEGF [74]. Direct evidence also suggests that M2 macrophages produce cytokines, such as CCL18, which support EMT and the invasion and migration of tumor cells [75]. Cytokines are products of inflammation, and a close interaction among the production of pro-inflammatory factors in cancer cells, EMT, and the recruitment of inflammatory cells has been recognized. The main inflammation-related cytokines in this system are: the tumor necrosis factor (TNF)-α [76], interleukin-8 [77], and interferon (IFN)-γ [76]. These interactive events sustain each other and ally with each other to induce metastasis [73].

Taking the previously cited reports into consideration, it turns out that the activation of EMT in carcinoma cells can change their susceptibility to an immune attack. In murine mammary carcinoma models, tumors engaged with an epithelial differentiation profile were found to express high levels of MHC-I, low levels of PD-L1, and contain CD8^+^ T cells, together with M1 (antitumor) macrophages. In contrast, tumors with a mesenchymal gene expression pattern express low levels of MHC-I, high levels of PD-L1, and contain regulatory T cells, M2 (protumor) macrophages, and exhausted CD8^+^ T cells [78]. Mesenchymal trans-differentiated carcinoma cells might also protect epithelial tissue segments from immune attack. In lung adenocarcinomas with an EMT phenotype, the co-existence of immune activation with the elevation of multiple targetable immune checkpoint molecules, including PD-L1, PD-L2, PD-1, TIM-3, B7-H3, BTLA, and CTLA-4, together with an increase in their infiltration by CD4^+^Foxp3^+^ regulatory T cells, was observed [79]. The communication between EMT and PD-L1 expression in cancer cells, as was also found in other studies [44,45,80,81], is bidirectional, in which EMT transcription factors up-regulate the expression of PD-L1, while the latter, in turn, can promote the EMT process. This complex relationship between EMT and PD-L1 signaling plays an important role in the immune evasion of the tumors [80]. On the other hand, these findings may suggest that cisplatin-resistant EMT tumor cells could display an increased sensitivity to anti-PD-1 or anti-PD-L1 antibody treatment [82]. However, recent studies investigating gene expression signatures, predicting the efficacy of PD-1/PD-L1 inhibitors in different cancer types, found that tumors with EMT or the mesenchymal phenotype showed a lower response rate [83,84,85]. Besides the potential application of the evaluation of the EMT status as a predictive marker for checkpoint agents, these results suggest that therapies aiming at the reversal of the EMT process may potentiate the efficacy of this type of cancer immunotherapy [86,87]. In addition to T-cell-based immune responses, EMT has also been implicated in influencing tumor cell sensitivity to macrophages and natural killer cells [88]. As illustrated above, there is a close alliance of a tumor-supporting, frequently inflammatory stroma with tumor cell nests consisting of EMT cells. In fact, as indicated by Chargari and colleagues, originally, most host cells in the tumor stroma possess tumor-suppressing abilities, but, during a long co-existence with a developing progressive carcinoma, the stroma will be remodeled into a constructive helper, which promotes growth, invasion, and cancer metastasis [89]. A major player in this constructive cooperation of tumor stroma with a progressive cancer is the fibroblast component [89]. Indeed, fibroblasts (Figure 3) are involved in the activation of autocrine and paracrine molecular signaling pathways, which regulate tumor cell proliferation [38], cell death, responses to hypoxia, DNA repair, and EMT. Taking HNSCC as an example, fibroblasts and tumor cells are the main components of the tumor tissue, as depicted in Figure 3 [38].

The fibroblastic cell population, which has the most intimate interaction with the tumor cells undergoing EMT, consists of cancer-associated fibroblasts (CAFs). These cells interact with tumor cells and support their proliferative and invasive behavior [90]. These fibroblasts allow tumor cells to undergo EMT through the secretion of cytokines, such as interleukin-6 and the main EMT inducer, TGF-β1 [20,24,38,91,92]. TGF-β1 is produced by cancer cells and, particularly, by stromal cells in an attempt to control inflammation. Interestingly, TGF-β1 can activate the resident fibroblasts to convert them into myofibroblasts by expressing alpha smooth muscle actin, which is associated with tumor invasion [45]. Some of these ‘myofibroblastic’ CAFs might derive from carcinoma cells undergoing EMT [37]. Interestingly, these CAFs that support EMT in tumor cells are reported to be senescent [93], whereas normal dermal fibroblasts are able to attenuate the growth of tumor cells, which has been reported in melanoma by Nwani et al. [94].

### 2.3. The Path from EMT to Radio/Chemoresistance

EMT plays important roles in cancer progression and metastasis. While the importance of EMT-TFs in metastasis is controversial [2,42], they are clearly related to cancer drug resistance [42]. This relation was already described in 1992, reporting a significant increase in vimentin expression in adriamycin-resistant MCF-7 cells, accompanied by a reduced formation of desmosomes and tight junctions [95]. Similar to the questions concerning the contribution of EMT-TFs to metastasis, the investigation of EMT and cancer drug resistance also benefits from the use of lineage-tracing models. A highly significant model for the EMT relation to drug resistance was the EMT lineage-tracing system, established by Fischer et al. [35]. They generated Fsp1 (fibroblast specific protein 1) promoter-driven Cre recombinase, which activated GFP expression upon lox cleavage. This tracing followed cancer cells in stages of mesenchymal and epithelial differentiation. They found that GFP-positive cells that express the fibroblast specific protein 1 were resistant to apoptosis under chemotherapy treatment [35]. Apoptosis resistance is one possible mechanism by which EMT contributes to cancer drug resistance [96]. A frequently discussed and important drug resistance mechanism is the excessive drug efflux caused by the ATP-binding cassette (ABC), which is a component of cell membrane transporter proteins. EMT cells overexpress the ABC transporters [97,98]. Moreover, the promoters of ABC transporters contain binding sites for EMT-TFs [99].

Several authors investigated how a radio-resistant or a chemo-resistant phenotype develops through a resistance-related gene expression pattern. For instance, OSCC cell lines resistant to cisplatin can be experimentally created from their parental counterparts. In this case, the resistant cells exhibit an enhanced proliferative, clonogenic capacity, with a significant up-regulation of P-glycoprotein (ABCB1), c-Myc, survivin, and beta-catenin and a putative cancer-stem-like signature, with an increased expression of CD44, whereas they display a loss of E-cadherin [100]. In this respect, two different but comparable issues can be seen in relation to EMT and therapy resistance. Several pieces of evidence revealed that, if EMT is experimentally induced in cultured epithelial carcinoma cells, they will display cisplatin-resistance and radio-resistance [25]. Moreover, if, in a pure epithelial cancer cell culture, radio-resistance is induced [101,102], the resulting gene expression profile will resemble an EMT phenotype. A novel hypothesis suggests that EMT cells release membranous vesicles with a diameter of 40 to 100 nm, called exosomes, which are seen as functional mediators of a tumor–stroma interaction and of EMT. The exosomes are supposed to promote environmentally-mediated therapy resistance [103]. The transcription profile induced by EMT acquires radio-resistance in tumor cells that share the properties of stem cells [104].

As mentioned previously, both in solid tumors and in leukemia, the tumor or bone marrow microenvironment contributes significantly to EMT-related therapy resistance. Microenvironmental components and infiltrating inflammatory cells are able to provide mediators, such as TGF-β1 [24] neurotrophins, as brain-derived neurotrophic factor [20,23,105], which transform epithelial cancer cells into EMT cells. A stress-responsive gene, tissue transglutaminase 2 (TG2), is frequently up-regulated during inflammation and wounding, and this gene was reported to be aberrantly up-regulated in multiple cancer cell types. In breast cancer, Agnihotri, Kumar, and Mehta found that TG2 existed in tumor cells selected for resistance to chemotherapy and radiation therapy [106]. Mechanistically, TG2 is associated with the activation of the nuclear transcription factor-kappa B (NF-κB), Akt, focal adhesion kinase (FAK), and a hypoxia-inducible factor (HIF) [106].

One of the main contributions of fibroblasts to tumor progression is supporting the radio-resistance of tumor cells [25,38]. This is partly due to the induced peritumoral desmoplastic reaction [89]. A particularly interesting component, and rarely investigated part, is the behavior of the irradiated stroma. This is the component that recruits bone marrow-derived progenitors and allows for neovascularization, which seems to be a regeneration process, but, in fact, it is required for tumor relapse, after radiotherapy [89]. The tumor relapse can be explained by the mesenchymal transdifferentiated subpopulations of cancer cell nests that are induced by the fibroblastic stroma and are readily radio-resistant and chemo-resistant [106]. This EMT trans-differentiation pathway is achieved through STAT3 and Slug [104]. A sequential set of chemotherapy treatments or a series of irradiation treatments of pure epithelial cancer cell cultures induce resistant cells that also share the EMT phenotype [101,102,104,107]. These cells were not contacted by any external humoral factors or co-cultured with mesenchymal cells before. Considering this knowledge, one might conclude that EMT, which is originally a transcription-induced phenotype and differentiation switch, achieves its clinical significance in shared properties with therapy resistance. Additionally, EMT-regulating mechanisms seem to be relevant and important targetable mechanisms for enabling therapy success in refractory cancer tissue.

A unique property shared between the therapy resistant phenotype and EMT is the induced senescence in tumor cells [107]. In cultured colon cancer cells, senescence could be induced by exposure to 5-FU. These cells exhibited a senescence-associated secretome, which was capable of a paracrine induction of EMT in colon and rectal cancer cell lines, and it increased cell invasion in vitro [107]. Moreover, Tato-Costa et al. used tissue samples of rectal cancer patients treated with neoadjuvant radio-chemotherapy and found increased mRNA expression levels of EMT-related proteins (Slug, Snail, and vimentin) in tumor cell niches enriched for senescent cells [107]. Senescence might be a response of tumor cells both to microenvironment-induced stress, as well as to therapy. A further, frequently discussed environmental condition is hypoxia, which is also a niche, where EMT cells reside both in solid tumors and in leukemia. Hypoxia is originally a deadly microenvironment, but it allows a selection of tumor cells to be capable of inducing survival signaling pathways and spread in the tumor microenvironment. A broad distribution of hypoxia-related markers among tumor cells is a negative prognostic factor for treatment success. This survival challenge is clearly related to the induction of an EMT profile [108,109]. Hypoxia has shared signal pathways with EMT and provides tumor cells with a stem cell-like property [108]. A possible morphological endpoint of the torture of tumor cells by environmental and therapeutic circumstances is the acquisition of the polyploid giant cancer cells (PGCCs) (Figure 4) [110], which are a morphologically distinct subgroup of human tumor cells, with an increased nuclear size or multiple nuclei. These cells have been thoroughly investigated by Zhang et al. They described a profile of differentially regulated proteins in giant cells in response to hypoxia, stem cell generation, chromatin remodeling, cell cycle regulation, invasion, and metastasis [110]. The up-regulation of HIF-1α and its known target, stanniocalcin-1 (STC1), are common in PGCCs. The findings of Zhang et al. allow for the conclusion that a panel of stem cell-regulating factors and EMT regulatory transcription factors are up-regulated in PGCCs [110].

Hypoxia and the activation of HIF-1α have been reported to induce drug resistance by increasing the gene expression of MDR1 [109].

The previously summarized data revealed that either repeated therapeutic challenges or hostile environmental effects cumulate stress on tumor cells, which respond to an EMT transcriptional program. If hypoxia is involved in the challenges, the cells also provide a stem cell-like survival profile. The therapy resistance contains elements of altered death receptor signaling [111], the induction of survival signaling pathways [111], the induction of DNA repair proteins [25], and cell division changes, such as senescence [107], instead of a cell division cycle, as well as the generation of giant cells by DNA replication without cell division [110]. The EMT therapy resistance and cancer stem cell profiles share the mechanisms of transcription regulation [89,100,107,110]. Several miRNAs are involved in these transcription regulatory pathways.

Cisplatin-resistant and recurrent oral squamous cell carcinoma cells contain differently regulated miRNAs, compared to primary OSCC. Compared to primary tumors, in recurrent OSCC, miR-130b, miR-134, and miR-146b showed a significant down-regulation, while miR-149 showed a significant up-regulation [100]. These data, produced by Ghosh et al., clearly indicate that the expression profile of microRNAs is a significant indicator of a therapy-resistant gene expression pattern [100]. This list can be extended with miR-3591-5p, which is the most significantly increased miRNA in response to radiation [102].

Evidence that is published and broadly available confirms that radio-therapy or chemotherapy resistance in cancer is based on a shared transcriptional profile of EMT. The question is whether the novel mechanistic insight into these shared gene expression regulatory pathways allows for a clinically usable therapeutic inhibition of EMT.

### 2.4. Therapeutic Targeting of the EMT Fueling Radio/Chemoresistance

The first aim in influencing EMT through therapeutic action is to inhibit its signaling pathways. As mentioned in the Introduction, the first reports on EMT by Elisabeth Hay described the involvement of WNT-signaling in EMT [11]. Drug targeting of the EMT pathway, with the potential of clinical utilization, would involve the translation of the mechanistic research data of EMT-founders into clinical praxis [49]. Therapeutic agents targeting the WNT-signaling pathway have already reached clinical trials, and the highest potential is expected from a porcupine (PORCN) inhibitor, LGK-974 (WNT-974). LGK-974, which is a pyridinyl acetamide derivative, is an orally administrated drug that inhibits the gene expression of WNT-related genes and the WNT-dependent phosphorylation of LRP6. In preclinical settings, LGK-974 inhibited the growth of pancreatic cancer and HNSCC cells [49]. LGK-974 is absorbed within three hours, and its elimination half-life is 5–8 h [112]. It is the object of a National Cancer Institute trial (NCT01351103) to find the recommended dose of LGK-974, as a single agent and in combination with PDR001, that can be safely given to adult patients with selected solid malignancies, for whom no effective standard treatment is available (source: National Cancer Institute, Bethesda, MD, USA).

In the EMT process, in inflammation-fueling tumor progression and also in CAFs, activated NF-κB alpha plays significant roles. NF-κB alpha is targeted by denosumab, which is a humanized monoclonal antibody to the receptor activator of the NF-κB ligand (RANKL), which reached phase III of an ongoing clinical trial, while evaluating its therapeutic effect (Table 1). This trial in breast cancer hypothesized a potential for interfering with bone metastasis or disease recurrence in patients with early-stage breast cancer, whereas denosumab, in a clinical trial including 4509 patients, did not improve disease-related outcomes [113].

The targeting of EMT-related signal transduction is supposed to induce the reversal of mesenchymal trans-differentiation. In a preclinical study, Jin et al. utilized simvastatin on experimentally developed radioresistant esophageal cancer cells. Simvastatin sensitized the radioresistant cells, suppressed their proliferation, migration, and invasion, and reversed their EMT. PTEN-PI3K/Akt were the main components in the effect of simvastatin [50]. In a further study, Zang et al. reported that STAT3 inhibition or Twist depletion reversed the EMT process and attenuated radio-resistance [51]. A similar concept, known as the inhibition of IL-6/STAT3 signaling, is also achieved by silibinin, which is a plant flavonoid that entered scientific focus in recent years, based on results that describe the possibility of radio-sensitizing endothelial cells by inhibiting the expression of pro-angiogenic factors [114]. Silibinin is a polyphenolic flavonoid isolated from milk thistle, which exhibited anti-neoplastic activity in both in vitro and in vivo cancer models, including a prostate cancer model (Table 1) [115]. Silibinin was experimentally combined with ionizing radiation, which contributed to the down-regulation of endothelial cell proliferation, clonogenicity, and a tube formation ability. It also synergistically reduced the migratory and invasive properties of prostate cancer cells [116]. Several pro-survival signaling pathways, considered to be the EMT’s primary activators, are inhibited by silibinin, such as Akt, Erk1/2, and STAT3 [116]. In this regard, silibinin has an appropriate targeting potential and candidates for an EMT-targeting drug with a good potential for radiotherapy sensitizing effects. Additional evidence has shown that silibinin is also instrumental in promoting therapy targeting immune checkpoint inhibitors. Alongside its impact on EMT regulators (e.g., SLUG, vimentin, and CD44), silibinin treatment significantly inhibited the up-regulation of the immune checkpoint regulator PD-L1 [117].

In prostate cancer, an extensive set of experimental data has led to silibinin clinical trials using a commercially-available silibinin formulation known as silybin-phytosome. In phase I, the clinical trial assessed the toxicity of a high-dose orally administered silybin-phytosome in prostate cancer patients, and the most prominent adverse event was hyperbilirubinemia, with grade 1–2 bilirubin elevations in 9 of the 13 patients. The only grade 3 toxicity observed was an elevation of alanine aminotransferase (ALT) in one patient. No grade 4 toxicity was noted. Taking the efficiency of the silibinin-containing phytosome into consideration, no objective PSA responses were observed [115]. Silibinin is currently in a phase II clinical trial for prostate cancer patients. In vitro results suggested that silibinin enhanced the radiotherapeutic response of prostate cancer cells by suppressing ionizing radiation-induced pro-survival signaling and DNA double strand break repair pathways [118]. The most recent results of the phase II trial show that high silibinin concentrations of oral silybin-phytosome can be achieved in blood transiently, but low levels of silibinin are seen in prostate tissue [119]. This could have been the reason for the lack of the PSA response in the previously mentioned trial [115]. Silibinin’s inability to penetrate tissue may be explained by its short half-life, the brief treatment duration [119], or a yet unknown active process that removes silibinin from the prostate [119]. The use of silibinin in the prostate cancer trial is an example of how the mechanistic knowledge of understanding the EMT mechanisms can provide compounds that are currently entering clinical use.

As discussed above, microRNAs play a significant role in regulating EMT, cancer stemness, and therapy resistance conditions. This knowledge allows for the hypothesis that microRNAs may also be suitable for influencing these processes to obtain a therapeutic advantage. Antoon et al., for example, identified miRNA alterations associated with the development of resistance to death receptor signaling [120]. They utilized the pharmacological inhibition of the p38 mitogen-activated protein kinase (MAPK) (RWJ67657), which resulted in a decreased tumor growth in xenograft models [120]. The inhibition of p38 partially reversed the EMT changes and led to a reduction of the gene expression of certain EMT markers, such as Twist, Snail, Slug, and ZEB. In addition, RWJ67657 treatment also altered the expression of several miRNAs, known to promote therapeutic resistance, including miR-303, miR-302, miR-199, and miR-328 [120].

There is a recent finding that the tumor suppressor, p21, does not only function as a cell cycle inhibitor, but it also plays a role as a transcriptional co-repressor [121]. MicroRNAs have been identified as p21-regulated targets. Three abundant miRNA clusters, miR-200b-200a-429, miR-200c-141, and miR-183-96-182, were induced by p21. If in a model system, p21 was lost, the miR-200 family was down-regulated, and EMT was induced in the corresponding cells [121].

In addition to the experimental results, microRNA-related transcription regulatory processes also reached clinical utilization levels, as synthesized mimics and modified miRNAs (Table 1).

MicroRNA-34 (miR-34) has been reported to be dysregulated in various human cancers and regarded as a tumor-suppressive microRNA because of its synergistic effect with the tumor suppressor, p53. The first miRNA-based clinical trial (NCT01829971) contains the tumor-targeted microRNA drug, MRX34, which is based on miR-34a mimics. The corresponding miRNA, MiR-34, might be an EMT repressor [122]. The aim of the available first phase I microRNA study assessed the maximum tolerated dose (MTD), safety, pharmacokinetics, and clinical activity of MRX34, a liposomal miR-34a mimic, in patients with advanced solid tumors. One patient with a hepatocellular carcinoma (HCC) achieved a prolonged confirmed partial response, lasting 48 weeks, and four patients experienced disease stabilization, which are the first clinical indications of the usability of an EMT-targeting miRNA concept. MRX34 treatment with dexamethasone premedication was associated with an acceptable safety level and showed evidence of antitumor activity in a subset of patients with refractory advanced solid tumors [123]. The utilization of either pharmacological EMT pathway inhibitors or stabilized EMT-modifying RNA products confirms that EMT research attracted interest at the clinical level. As presented in Table 1, some clinical trials use compounds that modify regulators, growth factors, and cytokines that induce, influence, or regulate EMT. These include antibodies, cytostatics, and small molecules. In 2014, Kothari, Ki, Zapf, and Kuo [124] reviewed the development of compounds that target EMT. A particularly interesting part of this review article concerns TGF-β1 targeting. One compound mentioned in the review, LY2157299, also reached clinical application and is in the phase I trial of Rodon et al. [125] (Table 1). A favorable toxicity profile and 15% disease control rate was reported in a refractory/relapsed malignant glioma. In addition to LY2157299, fresolimumab, which is an antibody that neutralizes all TGF-β isoforms, is also being evaluated in phase I/II clinical trials [49,126] (Table 1). As Kothari et al. [124] also discussed, metformin decreases key EMT-TFs, including ZEB1, Twist1, and SNAI [42,127,128], and may be considered an EMT-targeting drug. Arrieta et al., in 2019 [129], reported the results of a phase II clinical trial, considering a combination of metformin and EGFR tyrosine kinase inhibitors (Table 1) in advanced lung adenocarcinoma. The authors argue for a significantly improved progression-free survival through the use of metformin, combined with EGFR inhibitors (Table 1). A further compound mentioned by Kothari et al. [124] was curcumin. Curcumin-sensitized colorectal cancer cells were resistant to 5-fluorouracil through the miRNA-mediated suppression of EMT [130]. Our research group published one of the first reports on the preclinical effects of curcumin on EMT [22]. The main problem associated with the therapeutic use of this promising polyphenol was its very low availability due to a poor uptake, which was overcome by the development of the Theracurmin microparticle. Theracurmin was applied in a phase I clinical trial by Kanai et al. in 2013 [131] in pancreatic or biliary tract cancer patients. The trial reported a 25% disease control rate (DCR) with no unexpected adverse effects (Table 1).

Lastly, we would like to return to the Introduction of this review article. A major concept in EMT-targeting therapy is its reversal. This has been achieved by the histone deacetylase inhibitor, vorinostat [12,132]. In a novel phase I clinical trial by Teknos et al. [132], in stage III IVa and IVb head and neck cancer, vorinostat with concurrent chemoradiation achieved a 96.2% complete response rate (Table 1). 

## 3. Conclusions

EMT is a complex regulation process, which might be a stress management mechanism of epithelial tumor cells in solid tumors responding to a hostile microenvironment, acidification, or hypoxia. The same transcription regulatory process is also a response of epithelial tumor cells to experimentally repeated sequential chemotherapeutic treatments or to repeated ionizing radiation cycles. EMT sustains itself with inflammation in cancer tissue and contributes to the suppression of an effective antitumor immune response and to the induction of immune checkpoint regulators. The experimental data suggest that EMT is a reversible transcriptional process, which contains targetable elements, such as signaling junction points or regulatory miRNAs, and some of these mechanisms have already been targeted by compounds designed for clinical application.

## Figures and Tables

**Figure 1 cells-09-00428-f001:**
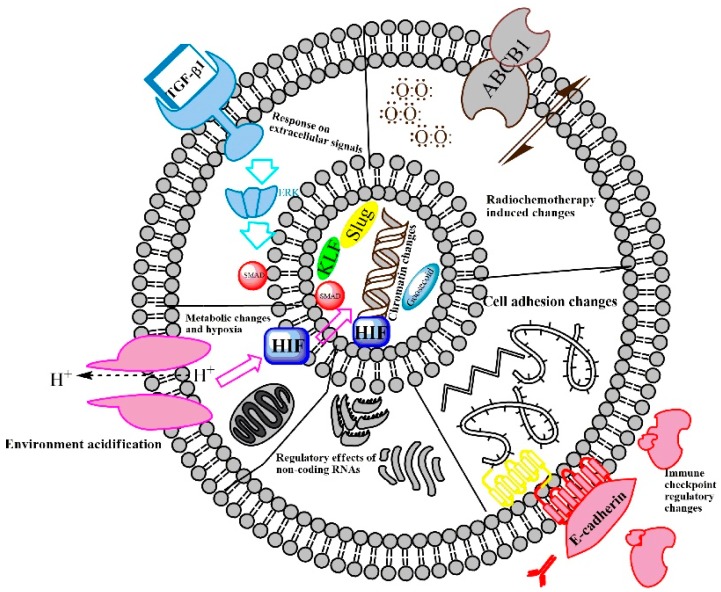
Main EMT regulatory mechanisms. EMT is a product of internal and external events in cells. It is mainly induced by external factors, such as TGF-β1, neurotrophins, or interleukin-6, which activate several signaling pathways such as WNT, Smads, STATs, Akt, or Erk1/2. The activated pathways drive transcription factors (Goosecoid, Snail, Slug, Twist, nuclear factor erythroid-2-related factor 2 (NRF2), and Krüppel-like factors (KLFs)) to bind to special responding sequences in DNA, and, consequently, these mainly suppress the expression of genes related to epithelial differentiation. The best-known event in this process is the down-regulation of E-cadherin by the SNAI transcription factors, Snail, and Slug. At the same time, mesenchymal gene products, such as vimentin, are induced during the transcriptional changes in the EMT process, where the Smad, HIF-1α, and KLF4 transcription factors are directly involved [43,44]. The transcription regulatory process during EMT does not only rely on transcription factors, but also on long noncoding RNAs (lncRNAs) and microRNAs. The external EMT-inducing factors, such as TGF-β1, neurotrophins, or interleukin-6, are available from the fibroblastic and inflammatory microenvironment, but tumor cells also actively participate in the completion of an EMT program [45]. Based on the experimental data of several research groups, including ours, it has been found that acidic extracellular conditions and, in a further step, a low oxygen tension (hypoxia) are necessary for EMT. Interestingly, EMT tumor cells produce immune checkpoint regulatory molecules and actively remodel their immune microenvironment into a more tumor-friendly one from a hostile neighborhood.

**Figure 2 cells-09-00428-f002:**
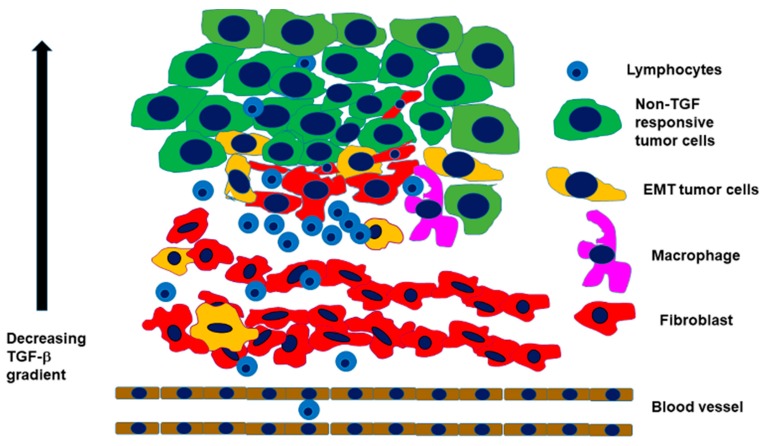
The TGF-β1 gradient causes heterogeneity in the tumor cell nests. There are higher TGF-β1 concentrations near the tumor vasculature, and the tumor cell nest has a decreasing gradient from its border towards the middle, which generates heterogeneity between the tumor–stroma interface and the tumor cell nest center [47]. The TGF-β1 gradient contributes to the heterogenic architecture of the tumor cell nest, where the invasive front has a completely different functional and differentiation profile than the core of the tumor cell nest. Additionally, at the invasive front, tumor cells interact with fibroblasts, leukocytes, and other stromal cells. Some scattered tumor cells undergo EMT at the tumor cell nest–stroma interface and contain both epithelial and mesenchymal markers. The figure is based on a figure published by Mayorca-Giuliani and Erler [48], which has been updated by the published data of Nieto et al. [29].

**Figure 3 cells-09-00428-f003:**
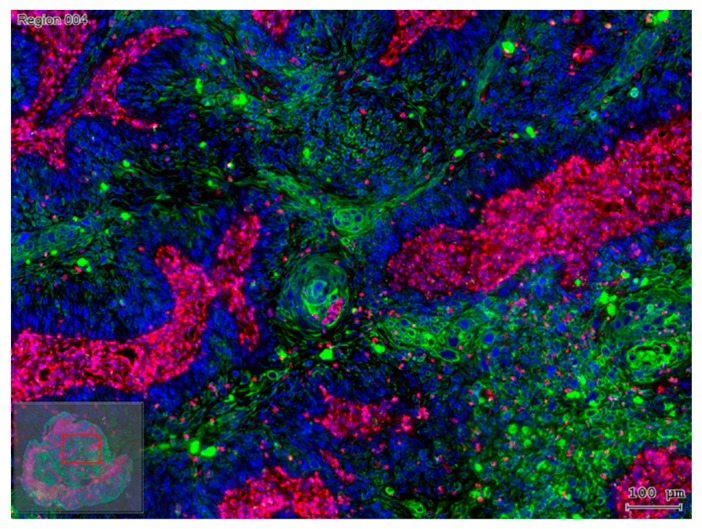
Vimentin represents an extended fibroblastic component in a primary oral squamous cell carcinoma (OSCC). Tissue section of human OSCC with the detection of vimentin (magenta) and pan-cytokeratin (green) and cell nuclei counterstained with DAPI (blue). Tumor cells form nests with a decreasing representation of the epithelial differentiation marker pan-cytokeratin from the middle of the tumor cell nests to their border. Among the tumor cell nests, the stroma contains large amounts of vimentin^+^ fibroblasts. This image was taken by the TissueFaxs^TM^ system (Tissuegnostics Vienna), and the bar represents 100 µm.

**Figure 4 cells-09-00428-f004:**
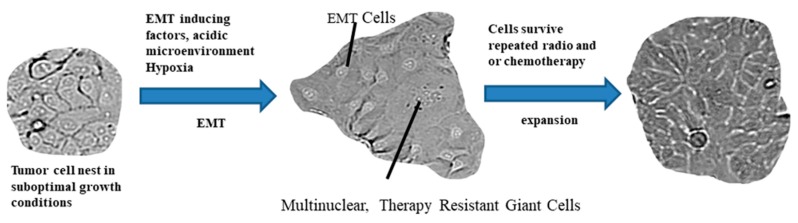
Multinuclear giant cells are therapy surviving intermediates. Hostile environmental conditions and hypoxia, in concert with EMT triggering factors, might induce EMT cells and polyploid giant cancer cells (PGCCs). These cells are also produced by serum-free culture conditions using conditioned media of oral squamous cell carcinoma cells and fibroblasts at pH = 6.7. These giant cells are also frequently the surviving cells of cultured OSCC cell lines during cisplatin or mitomycin C treatments. After the treatment cycles, the giant cells contribute to the expansion of the cell culture of normal epithelial tumor cell nests.

**Table 1 cells-09-00428-t001:** Examples of clinical trials targeting EMT.

Trial ID	Compound	Compound Type	Phase	EMT Target	Tumor Type	Side Effects	Outcome	Comments	Reference
NCT00487721	silibinin	flavonoid	I	signal transduction	Prostate cancer	elevation of ALT	no OR	low tissue penetration	Flaig et al. 2010 [119]
NCT01077154	denosumab	monoclonal antibody	III	NF-κB-alpha	breast cancer	neutropenia, leukopenia	no improvement		Coleman et al. [113]
NCT01829971	MRX34	miRNA	II	miRNA-related regulation	solid tumors refractory to standard treatment	fatigue, back pain, diarrhea, lymphopenia, neutropenia	PR, SD	dexamethasone pretreatment required	Zhang, Liao Tang 2019 [122]
NCT02536469	BMS-986253	monoclonal antibody	II	tumor microenvironment	locally advanced solid tumors	fatigue, hypophosphatemia hypersomnia	SD	safe, well tolerated	Chan et al. 2017 [133]
EudraCT 2012-004956-12	eribulin mesylate	taxane-based cytostatics	I	mesenchymal expression	primary triple-negative breast cancer	neutropenia	pCR	biological activity in triple-negative breast cancer	Di Cosimo et al. 2019 [134]
	LY3164530	bispecific antibody	I	inhibition of EMT in tumor cells	patients with advanced or metastatic cancer	rash, acneiform dermatitis, hypomagnesemia, paronychia, fatigue, skin fissures, hypokalemia	ORR: 10.3%, DCR: 57.7%	significant toxicities associated with EGFR inhibition	Patnaik et al. 2018 [135]
	LY2157299	TGF-β receptor kinase inhibitor	I	TGF-β1	refractory/relapsed malignant glioma		DCR: 15%	favorable toxicity profile and no cardiac adverse events	Rodon et al. 2015 [125]
NCT00356460	fresolimumab	monoclonal antibody	I	TGF-β1	advanced malignant melanoma and renal carcinoma	keratoacanthomas, hyperkeratosis	PR: 3%	no dose limiting toxicity	Morris et al. [126]
NCT03071705	metformin hydrochloride	biguanide	II	modifies EMT machinery	advanced lung adenocarcinoma		median PFS and OS significantly longer		Arrieta et al. 2019 [129]
ID: 000002950	Theracurmin	polyphenol	I	SNAIL	pancreatic or biliary tract cancer patients	no unexpected adverse events	DCR: 25%	significantly available curcumin	Kanai et al. 2013 [131]
NCT01064921	vorinostat	cytostatics	I	reversal of EMT	stage III, IVa, IVb HNSCC	anemia, leukopenia, lymphopenia	complete response 96.2%	with concurrent chemoradiation, safe and effective	Teknos et al. 2019 [132]

Abbreviations: MTD: maximum tolerated dose. SD: stable disease. pCR: pathological complete response. PFS: progression-free survival. DCR: disease control rate. PR: partial response. OR: objective response. ORR: objective response rate. OS: overall survival.

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
