# Peer review of "Epithelial to Mesenchymal Transition: A Mechanism that Fuels Cancer Radio/Chemoresistance"

_cells, 2020, doi:10.3390/cells9020428_

Round 1
Reviewer 1 Report
In this review the authors summarized the mechanisms of cancer cells epithelial to mesenchymal transition (EMT), and cross talk between EMT cells and immune cells. The authors review particularly the involvement of EMT process in resistance to chemotherapy and radiotherapy, and highlighted a very broad subject represented by the influence of EMT on cancer-radio-chemoresistance.
The review article gives some interesting informations, but it contains several weakness.
Therefore, the paper needs many improvements before publication.
First of all, the manuscript suffers from the usage of a poor English style, and a careful revision of grammar, syntax is highly recommended
The introduction is confusing. Moreover, the authors should explain and describe the morphological alterations leading to EMT.
The authors should eliminate the subsection 2 “Matherials and Method” which should be a part of Introduction section.
The authors should also delete the title “Results and Discussion” of the subsection 3.
In the abstract the authors mentioned several microenvironmental factors, such as fibroblasts and myofibroblasts, inflammatory, immune and endothelial cells are concerned with the induction of EMT in tumour cells. In the subsection 3.2 “Interaction with the inflammatory microenvironment “ the authors reviewed only the cross talk between cancer cells and inflammatory stromal cells.Therefore, the authors should also highlight the role of other stromal cells, such as fibroblasts and cancer associated fibroblasts (CAFs) in EMT regulation of cancer cells.
In particular fibroblasts can modulate differently cancer development. Normal dermal fibroblasts can block melanoma formation during its early stage. On the contrary senescent fibroblasts sustain melanoma growth.
In the subsection 3.3” The path from EMT to radio- chemoresistance” the authors should better discuss and analyse the mechanisms of EMT-induced radio and drug resistance (PMID: 27455225).
The authors should change the title of subsection 3.4 “The understanding of EMT fueling radio-chemoresistance – does it contribute to improvement of cancer treatment?”
In the subsection 3.4 the authors discussed the use of silibinin, in cancer strategies, combined with ionizing radiation. There are many drugs targeting EMT in clinical trials (PMID:31552191). Therefore, the authors should check for other papers analysing the use of molecules combined with ionizing radiation in cancer therapeutic strategies (PMID: 29208461; PMID: 28061440).
Author Response
Please find the attached file for Answers to the comments of Reviewer 1.

Reviewer 2 Report
The authors have written a good review however at times I am missing some depth and also some key references. Please see my specific comments below.
Introduction
In the introduction it would be good to mention Elisabeth Hay as she is the true founding member of the EMT field as well as other important contributors including Nieto and Weinberg. I found the referenced work not accurately highlighted the work of these authors and their prominent work.
EMT regulating mechanisms
Here the authors focus on an old EMT paper which are questioning the filed of EMT. The EMT field has been challenged not only by David Tarin in 2005 and it would be good to highlight more recent literature either in this or another section.
The authors are in the introduction discussing the role of EMT in leukaemia however very little is discussed about this subject matter later in the text.
According to Papiewska-Pajak and colleagues, Snail plays a critical role in inhibition of E-cadherin gene expression. Snail directly binds to responsive E-box sequences in the E-cadherin (CDH1) promoter. I am not following this reference the major discovery of Snail being a repressor of E-cadherin was made by Nieto and Herreros groups so I would advise referencing this statement.
A novel hypothesis suggests that especially EMT cells release membranous vesicles 40 to 100 nm in diameter, called exosomes, which are seen as functional mediators of tumor-stroma interaction and of EMT. The exosomes are supposed to promote environment-mediated therapy resistance. Again the referencing to this section is strange.
Author Response
Please find the attached file for Answers to the comments of Reviewer 2.

Round 2
Reviewer 1 Report
The paper has been substantially improved by the authors.